# Chromogenic, Biochemical and Proteomic Identification of Yeast and Yeast-like Microorganisms Isolated from Clinical Samples from Animals of Costa Rica

**DOI:** 10.3390/jof10030218

**Published:** 2024-03-16

**Authors:** Alejandra Calderón-Hernández, Nelly Castro-Bonilla, Mariamalia Cob-Delgado

**Affiliations:** 1Mycology Laboratory, School of Veterinary Medicine, Universidad Nacional, Heredia 40104, Costa Rica; 2Master Program in Tropical Diseases, Regional Graduate Program in Tropical Veterinary Science, School of Veterinary Medicine, Universidad Nacional, Heredia 40104, Costa Rica; ncastrob@ccss.sa.cr; 3Veterinary Teaching Hospital Program, School of Veterinary Medicine, Universidad Nacional, Heredia 40104, Costa Rica; 4Centro Integrado de Salud de Coronado, Caja Costarricense del Seguro Social, San José 11103, Costa Rica; 5Reference Laboratory in Mycology, Instituto Costarricense de Investigación y Enseñanza en Nutrición y Salud, Cartago 30301, Costa Rica; mcob@inciensa.sa.cr

**Keywords:** proteomic identification, pathogenic fungi, biochemical profile, chromogenic media, *Prototheca*, wild animals, domestic animals, veterinary mycoses, One Health

## Abstract

Yeast infections are challenging human and animal medicine due to low rates of detection and the emergence of unknown ecology isolates. The aim of this study was to verify the biochemical identification of yeasts and yeast-like microorganisms obtained from animals comparing the results with chromogenic media and matrix-assisted laser desorption/ionization time-of-flight (MALDI-TOF MS). Between January and August 2023, yeast and yeast-like isolates from samples of animals with suspicion of mycosis were identified using Vitek^®^ 2 Compact, Brilliance^®^ Candida Agar and MALDI Biotyper^®^ MSP. A total of 39 cases were included, and 45 isolations were obtained. *Cryptococcus neoformans* (15.5%, 7/45), *Meyerozyma guilliermondii* (13.3%, 6/45), *Candida parapsilosis* (11.1%, 5/45), *Candida albicans* and *Candida tropicalis* (8.9%, each one 4/45) were the most identified organisms. There was full agreement with the three identification methods in 71.1% (32/45) of the isolates, disagreement on species in 17.8% (8/45), disagreement on genus and species in 6.7% (3/45) and, in 4.4% (2/45), there was no matched pattern in MALDI-TOF to compare the results. Biochemical methods are a good option in laboratories where proteomics are not available, and chromogenic media enhances diagnostics by detecting mixed infections. Surveillance must be implemented to improve the detection of agents shared between humans and animals.

## 1. Introduction

Mycoses caused by yeasts are a growing challenge in human medicine, particularly in patients with immunosuppression, hospitalized, transplanted, carriers of medical devices or central catheters and with chronic diseases such as diabetes, mainly because its detection is less than 50% in cases of invasive mycoses or fungemia [1,2]. During the period of 2010–2019, reports on yeast infections doubled the publications available in the decade of 1990–1999 (50,000 reports against 25,000) but continues to be less inferior than bacterial articles (350,000) [3]. Yeasts infections are mostly caused by species of the genus *Candida* but also by emerging pathogens. The management of emerging pathogens are more difficult, because the ecology and pathogenicity are unknown, and if intrinsic antifungal resistance exists, then evidence-based treatment cannot be performed as expected, and failure can occur [1,4].

Regarding animals, yeast infections are rarely reported, except for those caused by the genera *Malassezia*, *Candida* and *Cryptococcus,* which cause otitis and dermatitis; subcutaneous masses or gastrointestinal, respiratory and central nervous system compromises [5,6,7,8]. The diagnosis of yeasts represents a challenge in veterinary medicine, since animals have closer contact with the environment and it is possible to find transitory or pathogenic environmental microorganisms in them [6,9,10]. These yeasts are usually identified by commercial biochemical assays designed to detect the most common human pathogens, so the databases are very limited, and the identification is given at the genus level, and to attempt the species, molecular identification must be performed [4,11]. As molecular methods are expensive, most times, they are not pursued by the guardians of the animals, and consequently, the identification is not carried out or is carried out later for research purposes.

To support the morphological and biochemical results, chromogenic culture media are a good resource to detect multiple agents in the primary culture to give a preliminary characterization of the isolates and to confirm the purity of a strain. Those kinds of media use chromogenic substrates, which, in the presence of certain enzymes, result in the formation of colonies of a particular color: blue (*Candida tropicalis*), green (*Candida albicans* and *Candida dubliniensis*), pink (*Pichia kudriavzevii*, formerly *Candida krusei*) and beige or brown (other yeasts) [12,13,14]. Nevertheless, interpreting the results is dependent on the brand of the media and the available images to compare the spectra of colors of the isolates that are available only for the most frequent agents on the web or in research papers [12,13,14].

An alternative for yeast identification (in centers where it is available) is matrix-assisted laser desorption/ionization time-of-flight mass spectrometry (MALDI-TOF MS). This is a simple, fast, specific technique that can identify several genera and species [15]. This proteomic system generates a spectrum of the proteins of the microorganism, which is then compared to those found in the database of the equipment used, thus obtaining identification. However, to obtain better results, the database must be enhanced with local isolates from multiple samples [16].

This research was carried out with the objective of verifying the identification of yeast and yeast-like microorganisms isolated from animals by comparing the results of a biochemical semi-automated method (Vitek^®^ 2 Compact) with a chromogenic media (Brilliance^®^ Candida Agar) and mass spectrophotometry (MALDI Biotyper^®^ MSP).

## 2. Materials and Methods

During January to August 2023, a survey on the identification of yeast and yeast-like isolates obtained from clinical samples of domestic and wild animals with suspicion of superficial and deep mycosis was made at the Mycology Laboratory of Universidad Nacional, Heredia, Costa Rica. Isolates stored in the laboratory collection and those obtained during the period were considered.

The samples were cultured by the streak method (in the case of swabs and liquids) or by small portions (in the case of biopsies and scrapings) on Sabouraud Dextrose agar (SDA) (Oxoid Ltd., Basingstoke, Hants, UK) and incubated at 28 °C and 37 °C for one week, apart from skin scrapings where it grew for two weeks and biopsies for three weeks. When it was available, primary cultures on Brilliance^®^ Candida Agar (BCA) (Oxoid Ltd., Basingstoke, Hants, UK) were made to detect mixed cultures.

Significant growth was considered when (1) colonies appeared over the inoculated sample in the case of biopsies and scrapings in at least one zone of growth using the streak method and when it was in the presence of encapsulated, budding or phagocytized yeasts or hyphae and pseudomycelia in the direct examination (Gram and/or Giemsa stains); (2) when the isolation came from a sterile organ; (3) when it was a culture of treatment monitoring of a previous diagnosed disease and (4) when it was a isolate submitted for identification from a private veterinary diagnostic laboratory.

Transitory growth was annotated when no fungal elements were seen in the direct examination. The significance of growth was not determined in the cases where no direct examination was made to classify it as significant or transitory. To confirm the purity and corroborate the microscopic morphology of the colonies, Gram staining was performed on each isolate. Gram staining was chosen to detect and classify the bacterial contamination of the isolates. Subcultures on BCA and SDA incubated at 28 °C for 24, 48 and 72 h were done to evaluate the macroscopic appearance. As a control for BCA growth and color development, the strains *Issatchenkia orientalis* derived from ATCC^®^ 6258, *Candida albicans* derived from ATCC^®^ 14053 and *Candida parapsilosis* derived from ATCC^®^ 22019 were used.

### 2.1. Biochemical Identification

Biochemical identification was made with Vitek^®^ 2 Compact (bioMérieux, Inc., Durham, NC, USA) using the yeast identification cards (YST), which allow the identification of fungi and fungal-like algae of the genera *Candida*, *Cryptococcus*, *Malassezia*, *Prototheca*, *Rhodotorula* and *Trichosporon* and the species *Geotrichum klebahnii*. To carry out the identification of the microorganisms, pure cultures on SDA after 24 h of growth were used (except for isolates of *Malassezia*, *Prototheca*, *Trichosporon*, *Cryptococcus* and *Geotrichum*, which were left up to 72 h). The incubation was in an aerobic atmosphere at a temperature of 28 °C, except *Malassezia* and *Prototheca* at 37 °C. The stored isolates were subcultured twice before this procedure. Solutions of the microorganisms were prepared in a sterile saline solution with a volume between 1.8 and 2.2 McFarland and reeded with Densichek (bioMérieux, Inc., Durham, NC, USA). Then, the recommendations of the commercial house were followed to handle samples using the equipment, and their interpretation was as follows: an excellent identification was considered if the probability was 96 to 99%, very good if the probability was 93 to 95%, good if the probability was 89 to 92% and, finally, it was an acceptable identification if the probability was between 85 and 88%.

### 2.2. Proteomic MALDI-TOF Identification

Proteomic identification by mass spectrophotometry with MALDI-TOF was performed using the MALDI Biotyper^®^ MSP (Bruker Daltonics GmBH & Co. KG, Bremen, Germany) from the Mycology Reference Laboratory of the Costa Rican Institute for Research and Teaching in Nutrition and Health (INCIENSA) to corroborate the biochemical identification. For this technique, cultures on SDA for 24 to 72 h incubated between 30 and 37 °C were used. A minimum amount of the colony of interest of the microorganism was placed with the help of a wooden stick on the 96-well metal plate, then 1 µL of 70% formic acid was added. It was allowed to dry, and 1 µL of the matrix was used. Once the plate was ready, the reading was carried out. Each sample was processed in duplicate, and *Escherichia coli* isolation was used as a calibrator and as a control. The identification at the species level was given if the first ten identities were the same microorganism and if at least the first three had scores above 1.75 [15].

### 2.3. Results Interpretation and Data Register

Macroscopic morphology in SDA and the microscopic appearance in Gram staining was compared to the clades given in reference literature [17,18]. The results of the BCA were analyzed following the prospect of the product and research papers [12,13,14]. When controversial results were obtained, the color of the colony in BCA was revised to support one of the results, and when no reference of the color was available, MALDI-TOF results were preferred over Vitek’s, because the nomenclature is actualized, the profile of proteins is more specific than biochemical reactions that are shared by multiple yeasts and MALDI-TOF can differentiate species from complex species.

The data were registered in Microsoft^®^ Excel^®^ (Microsoft 365 MSO, version 2401, 64 bits) sheets including the following information: laboratory identification; animal source; type of sample; microscopic direct examination results; classification of the growth (significant, transitory or significance not determined, as explained above); type of isolate (pure or mixed); photographs of the macroscopic morphology in SDA and BCA and of the microscopic morphology in Gram staining, Vitek^®^ 2 Compact identification with their percentage of probability and MALDI Biotyper^®^ MSP identification with their corresponding score.

## 3. Results

A total of 39 cases from 28 domestic and 11 wild animals were included in the study. Most of the cases were dogs 30.8% (12/39), followed by cats 28.2% (11/39), two-toed sloths (*Choloepus hoffmanni*) 12.9% (5/39) and mares 10.3% (4/39). The cases involved isolates from multiple samples, such as skin scrapings; swabs of the oral cavity and ear canal, esophagus and crop; uterine lavages and skin, lung, kidney, lymph node and colon biopsies, among others. Most of the cultures were 89.7% pure (35/39), and mixed cultures were also detected 10.3% (4/39). Significant growth was seen in 69.2% of the cases (27/39). Meanwhile, a quarter (25.7%) of the cultures were classified as transitory (10/39), and in two cases, it was not possible to determine the significance, as no microscopic direct examination was made (5.1%) (Table 1).

From those cases, 45 yeast and yeast-like isolates were obtained. The more frequent were *Cryptococcus neoformans* (15.5%, 7/45), *Meyerozyma guilliermondii* (13.3%, 6/45), *Candida parapsilosis* (11.1%, 5/45), *Candida albicans* and *Candida tropicalis* (8.9%, each one 4/45). There was full agreement with the three identification methods in 71.1% (32/45) of the isolates, partial agreement on the genus but different species in 17.8% (8/45), disagreement on the genus and species in 6.7% (3/45) and, in 4.4% (2/45), there was no matched pattern in the MALDI-TOF to compare with the Vitek^®^ 2 Compact results (Table 2).

All the isolates of *C. albicans* (4/4), *C. neoformans* (7/7), *N. glabrata* (2/2) and *M. pachydermatis* (2/2) had the same result in the biochemical and in the proteomic assays and their expected color in the chromogenic media. On the other hand, *M. guilliermondii* and *C. parapsilosis* were also well identified by both methods (5/6 and 5/5, respectively); however, MALDI-TOF could identify one *Candida orthopsilosis,* which Vitek identified as *C. parapsilosis*, and a misidentification of one *M. guilliermondii* as *Candida ciferri* occurred with Vitek. Despite that MALDI-TOF gave the spectra for *Prototheca zopfii* and *Prototheca wickerhamii,* both were identified only by Vitek, because those microorganisms were not in the database of MALDI-TOF (Table 2).

All the isolates of *C. neoformans* were from cats and significant (7/7); the majority of *M. guilliermondii* were transitory isolations from skin (5/6), and the other microorganisms were from distinct organs and animals classified as significant or transitory agents (Table 3).

The macroscopic morphology of the isolates on SDA were creamy colonies with a smooth or rough texture white, cream or fuchsia in color (Figure 1). Regarding the growth in BCA, the isolates ranged from cream, beige, orange, blue, green to purple colors (Figure 2). The microscopic morphology of the yeasts varied with respect to the identified genus, with round to oval multiple- or single-budding yeasts observed and the presence of hyphae and pseudohyphae (Figure 3). A detailed description of the isolates is given in Appendix A.

## 4. Discussion

Forty-five yeast and yeast-like microorganisms isolated from animals with signs of superficial and deep or systemic mycoses were identified in this study. Well-known medically important fungi such as *C*. *neoformans*, *C. parapsilosis*, *C. albicans* and *C. tropicalis* were frequently involved in the cases. Meanwhile, others less common but increasingly more so agents of infections, such as *M. guilliermondii*, *R. mucilaginosa*, *P. laurentii* and *G. candidum*, were also detected with significant or transitory participation, as well as fungi scarcely reported, such as *T. coremiiforme*, *C. jirovecii* and *D. nepalensis* and fungal-like achlorophyllous algae *Prototheca zopfii* and *Prototheca wickerhamii* [17,18,19].

Four agents detected here are included in the Fungal Priority Pathogens List of the World Health Organization (2022): *C. neoformans*, from the critical priority group and *N. glabrata*, *C. parapsilosis* and *C. tropicalis* from the high priority group [2]. The latter classification was made “to guide research, development and public health action”, focused on the listed fungi, and those pathogens were prioritized based on the capacity to cause invasive and systemic mycoses for which treatment and management challenges exist, as well as drug resistance, considering that, in Costa Rica, there are also few reports on yeast infections in humans, the majority retrospective studies of candidemia and cryptococcosis. Regarding candidemia, INCIENSA has the records from the period 2018–2021 of the blood cultures from the main hospitals, with *C. albicans*, *C. parapsilosis*, *C. tropicalis* and *N. glabrata* (in that order) the most frequently identified [20]. However, in the two largest hospitals of this country (Hospital San Juan de Dios and Hospital México), the order of the fungal pathogens varies between places and periods [21,22,23]. In reference to cryptococcosis, this disease is prevalent in the country, mainly in patients with Human Immunodeficiency Virus (HIV) (72.7%) but also in immunocompetent patients (20%) and the rest by drug-mediated immunodeficiencies [24]. All these fungal pathogens are transmitted by exposure to the environment or by nosocomial and medical devices contamination, which reinforces the necessity of collaboration between human and animal diagnoses to detect mutual sanitary threats.

*Cryptococcus neoformans*, the most frequently identified agent here (15.5%, 7/45), were only detected in cats, and in all the cases, its role was considered significant (Table 3). The affected organs included the nose, skin, lymph nodes and bladder, which is consistent with what has been reported in this animal: mycotic rhinitis with colonization of the nasal cavity, single or multiple skin lesions and lymph nodes enlargement are the main clinical presentation of cryptococcosis [5,25,26]. Due to the behavior of burying feces and hunting, cats are exposed to acquiring fungal agents from the environment, which can cause disease or remain on their surfaces as asymptomatic carriers, which is why, in the case of sapronosis (fungal infections acquired from the environment), they are good bioindicators of contaminated environments [27] and can indicate a common source of exposure for the people they live with. Cryptococcosis is the most common systemic fungal disease in domestic felines and manifests mainly as an inflammation of the nasal mucosa that may or may not spread to the lower respiratory tract and cause lymphadenopathy or infection of the central nervous system [27]. Contrary to what happens in humans, in which the disease caused by *C. neoformans* manifests mainly in immunosuppressed patients [18], in cats, it has been seen that this disease is also contracted by animals in which immunosuppression was not detected [27], opening up the possibility that a large inoculum from the environment can promote the development of signs.

Surprisingly, *M. guilliermondii* was the second-most isolated species (13.3%, 6/45), but its participation was classified as transitory from the skin in almost all the cultures (83.3%, 5/6) and, in only one case of a two-toed sloth (*Choloepus hoffmanni*) (INV-002-23), was considered significant. This fungus is an emerging cause of invasive infections in humans, and in animals, this yeast has been associated with integumentary and genitourinary tract infections in dogs and cats, alimentary tract infections in poultry, abortion and mastitis in cattle and fungal keratitis and abortion in horses [5,25,28] and has also been detected as a commensal of oropharynx and cloaca of rheas (*Rhea americana*) [29] and in the nasolacrimal duct in horses [30].

*Candida parapsilosis* is part of a complex with *C. orthopsilosis* and *C. metapsilosis,* which causes multiple infections in humans but is more relevant as causing endocarditis, and by its elevated minimal inhibitory concentrations (MICs) for echinocandins [17,18]. In animals, this complex causes the same clinics as *M. guilliermondii* [5]. Two species of this complex were identified in this study: *C. parapsilosis sensu stricto* was the third-most common agent (11.1%, 5/45), isolated mainly from superficial samples (80%, 4/5) from dogs and cats, but its detection was highlighted in a sample of prostate aspirate from a dog and in the ear canal from a bat. On the other hand, *C. orthopsilosis* was identified by proteomic from a uterine infection in a mare, and without MALDI, this species could not be determined.

Following with the cases of uterine infections in mares from this survey, apart from the mentioned isolation of *C. orthopsilosis*, 66.6% (2/3) were associated with *C. tropicalis.* Those findings are important for three reasons: (1) fungal endometritis in horses is rare, representing 1–5% of the reported cases [31]; (2) in a robust retrospective study of a 13-year period on equine uterine fungal isolates, yeasts were the most common fungal agents (69%, 70/102), but no information about the genera and species were given [32]; and (3) in a study with 453 samples from the reproductive tract of healthy mares, no isolates from those two agents were found, despite *Candida* spp. being the most frequent genus isolated (53.2%), suggesting they are not part of the normal microbiota [33].

*Geotrichum candidum* were also identified in a mare but in the skin, and the finding was considered significant. Even though it is a saprophytic fungus, it is not commonly isolated from the skin and hair of normal horses, and it has been associated with an extremely rare dermatitis in horses in Brazil, causing areas of alopecia and producing scaling lesions [34]. Also, a case of fatal diarrhea in a horse with systemic dissemination was also reported in Brazil [35], so veterinarians should be aware of this agent in countries with similar weather to Brazil in case it could be detected on some other occasion.

It is worth mentioning digestive system isolates from dogs in which *Trichosporon asahii* (66.7%, 2/3) and fungal-like algae *P. zopfii* (33.3%, 1/3) were detected. The first one is not related to gastrointestinal disorders in animals and, in humans, is considered a contaminant of that kind of samples but an important pathogen found in samples of blood, urine and bronchoalveolar lavages [17,18]. Here, it was considered significant, as phagocytosed and free yeasts were observed in the microscopic direct examination of both colon biopsy samples, and it was isolated as pure. Therefore, further investigation is needed to elucidate the role of this agent in canine colitis, since no similar reports were found. On the other hand, *P. zopfii* is an expectable agent, since it has been reported in similar cases in Costa Rica [36] but, in other countries, is not a common finding. *Prototheca zopfii* and *Prototheca wickerhamii* isolates were not identified by MALDI-TOF, as the genus was not included in the database, and the same situation occurred in research from Belgium with animal isolates [37].

Another important fact is that the two-toed sloth (*Choloepus hoffmanii*) was the third-most frequent animal (12.9%, 5/39) from which isolates were recovered, the majority being from the digestive system (60%, 3/5), and the involved agents included *C. albicans*, *M. guilliermondii*, *C. jirovecii* and *D. nepalensis*. Specimens of this herbivorous wild animal, the national symbol of Costa Rica, are commonly encountered in rescue centers by different causes, but gastric complications by stress and diet adaptations are the main complications in captivity [38]. Mycotic glossitis/stomatitis and gastritis have been reported in sloths by histopathology, suggesting candidiasis [39,40,41]; nevertheless, case reports involving identification of the fungi and studies on their mycobiota were not found.

Although reports of yeasts in wild animals are very scarce, their transient isolations in those kind of animals in the present study were not surprising, since, according to a review on the evolution and genetics of yeasts, of the 76 new species described recently, only seven were related to human environments, the other sources being soil, plants, fruits, water and insects, among others [42], and those other sources are more likely to get in touch (by contact or by ingestion) with wild animals than humans. Also, yeast passes the gastrointestinal tract and can be isolated from feces, like in a survey of 325 droppings of parrots in Italy, where 49.2% from those were positive to 27 different species of yeasts, the majority from the genus *Candida* but also from the genera *Debaryomyces*, *Geotrichum*, *Pichia* and *Rhodotorula* [43]. Another example is the research on the mycobiome of wild herbivorous mammals (genus *Neotoma*) in 25 populations in the Southwestern United States, in which fungal assemblages were dominated by plant pathogens and molds, but their compositions varied by the animals’ habitats and significantly decreased in captivity [44]. Therefore, in the cases where the significant isolation of yeasts was detected in this study, it could be attributed to the stress of captivity and antibiotic therapy to which animals are exposed.

Chromogenic media provided benefits to identified mixed cultures, particularly in samples from the digestive system from exotic and wild animals. In this survey, mixed cultures with two or three isolates were detected in an ulcerative skin lesion of a cat (4302-21S), the lung of a Finch’s parakeet (*Psittacara finschi*) (INV-006-22), the crop of a southern mealy parrot (*Amazona farinosa*) (INV-009-22) and the oral cavity of a two-toed sloth (*Choloepus hoffmanii*) (INV-010-22). Also, the results of the BCA helped to identify isolates with controversial results in Vitek. One of the cases was VS-382-22, where the Vitek probability was *Candida ciferri* (98%) and the MALDI result was *M. guilliermondii* with a score of 1.88; despite no data on BCA growth for *C. ciferri*, all the isolates of *M. guilliermondii* were purple, as well this isolate (Figure 2J). Another controversial case was INV-010-22, where *C. tropicalis* (91%) was identified by Vitek, and the score for *C. jirovecii* was 1.92 as the result of MALDI. Both species have similar morphologies in SDA (Figure 3H,M), and by their hexosaminidase activity, both species could have greenish-blue appearance in BCA [12,13] (Figure 2M). Microscopically, they could be differentiated by the presence of arthroconidia in *C. jirovecii* (Figure 3M), but in this case, molecular identification is needed corroborate the species.

Biochemical identification by the Vitek^®^ 2 System was a good method to identify yeasts from animal origins, especially when the isolate came from a domestic animal and *C. albicans*, *C. neoformans* and *N. glabrata* were suspected, since, in the present research, all the isolates from those species had excellent, very good and good probability with this biochemical method, and MALDI-TOF corroborated their identity with scores higher than 1.7.

MALDI-TOF is routinely used in the diagnosis and research of pathogens in humans [45], but with veterinary isolates, it has been more focused on isolates from foods of animal origins [46] or detecting contaminants from animal excreta [47] and for research purposes [37] rather than for diagnosis. In this research, MALDI-TOF was a fair method to correct species with similar biochemical profiles given by Vitek, like *C. ciferii* vs. *M. gulliermondii*, *G. klebahanii* vs. *G. candidum*, *R. glutinis* vs. *R. mucilaginosa* and *Trichosporon* sp. [18], and to also detect complex species, like in the case of *C. orthopsilosis*.

## 5. Conclusions

The relationship between humans and animals is increasingly closer and, with it, the transfer of potentially pathogenic microorganisms between them. For that reason, medical mycology procedures must be unified between human and animal diagnoses, and databases must be available for global consultation and enlarged with isolates obtained from animals and the environment to improve the detection of emerging pathogens coming from those sources. Finaly, multidisciplinary works involving government, academia and health practitioners are necessary to detect threats to public health and generate useful information for One Health purposes.

## Figures and Tables

**Figure 1 jof-10-00218-f001:**
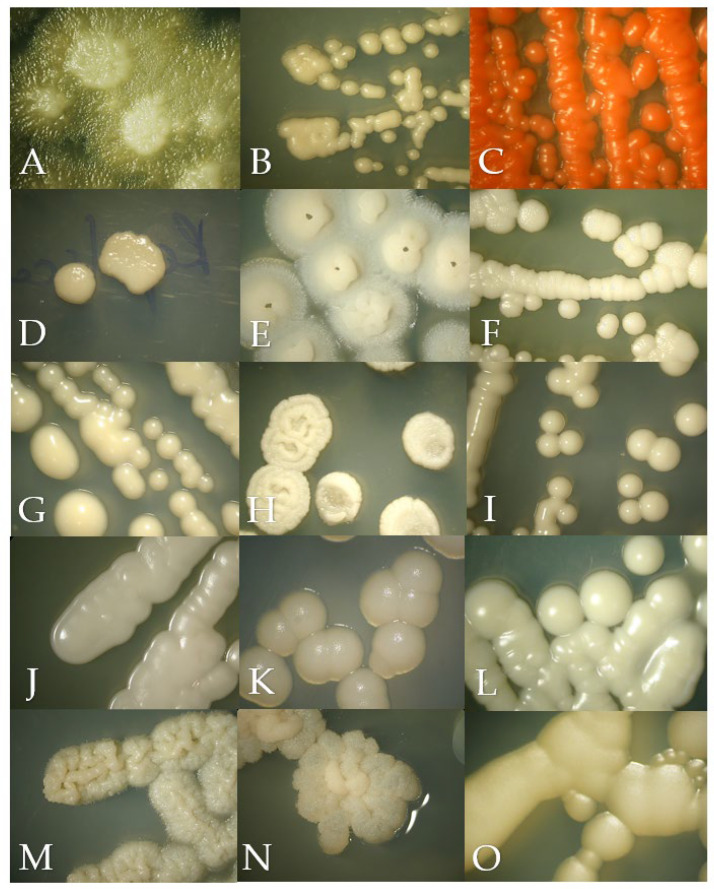
Macroscopic morphology of the isolates in Sabouraud Dextrose Agar after 24 to 72 h of incubation at 28 °C, except for *Malassezia pachydermatis* incubated at 37 °C. (**A**) *Geotrichum candidum*, (**B**) *Cryptococcus neoformans*, (**C**) *Rhodotorula mucilaginosa*, (**D**) *Prototheca zopfii*, (**E**) *Trichosporon asahii*, (**F**) *Candida parapsilosis*, (**G**) *Nakaseomyces glabrata*, (**H**) *Candida tropicalis*, (**I**) *Candida albicans*, (**J**) *Meyerozyma guilliermondii*, (**K**) *Malassezia pachydermatis*, (**L**) *Papiliotrema laurentii*, (**M**) *Cutaneotrichosporon jirovecii*, (**N**) *Trichosporon coremiiforme* and (**O**) *Debaryomyces nepalensis*.

**Figure 2 jof-10-00218-f002:**
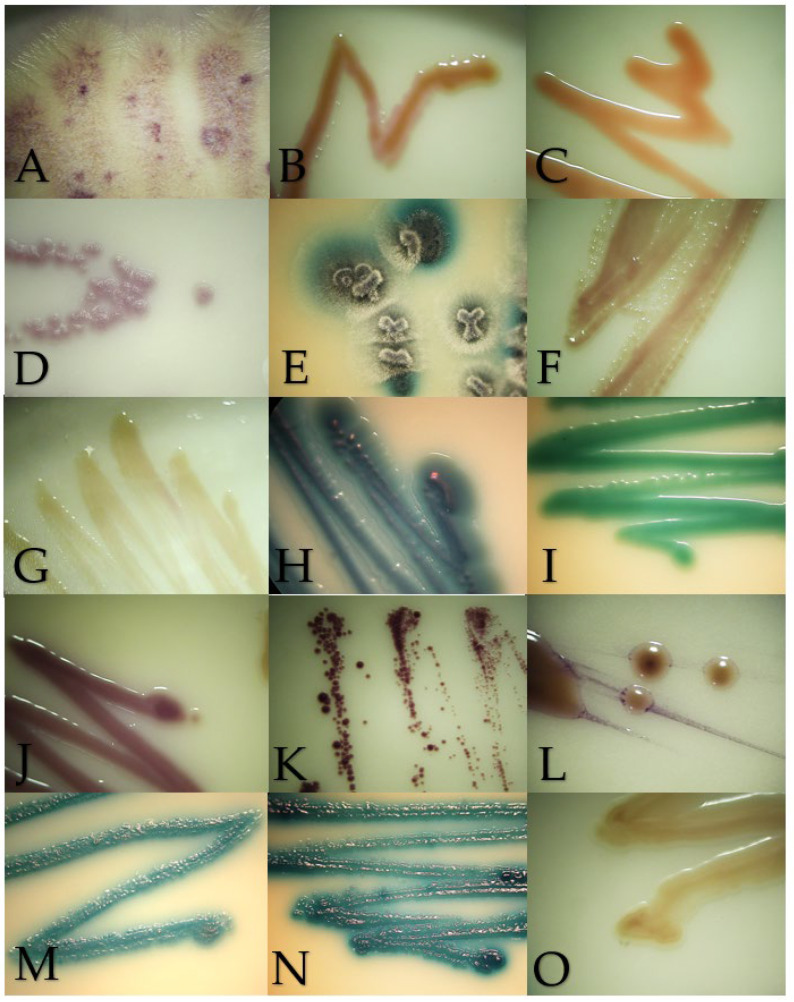
Macroscopic morphology of the isolates in Brilliance Candida Agar after 48–72 h of incubation at 28 °C, except for *Malassezia pachydermatis* incubated at 37 °C. (**A**) *Geotrichum candidum*, (**B**) *Cryptococcus neoformans*, (**C**) *Rhodotorula mucilaginosa*, (**D**) *Prototheca zopfii*, (**E**) *Trichosporon asahii*, (**F**) *Candida parapsilosis*, (**G**) *Nakaseomyces glabrata*, (**H**) *Candida tropicalis*, (**I**) *Candida albicans*, (**J**) *Meyerozyma guilliermondii*, (**K**) *Malassezia pachydermatis*, (**L**) *Papiliotrema laurentii*, (**M**) *Cutaneotrichosporon jirovecii*, (**N**) *Trichosporon coremiiforme* and (**O**) *Debaryomyces nepalensis*.

**Figure 3 jof-10-00218-f003:**
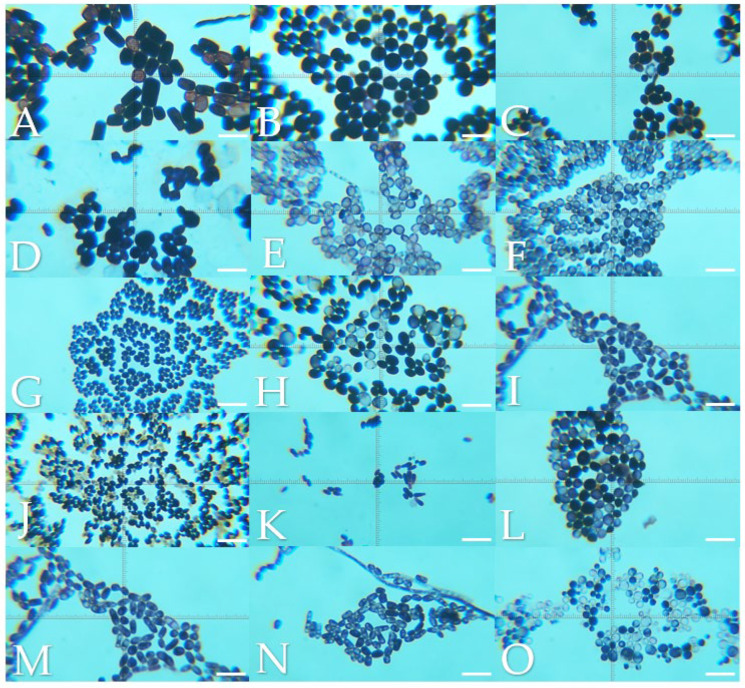
Microscopic morphology of the isolates by Gram staining in 1000× immersion oil. Scale bar 10.33 µm. (**A**) *Geotrichum candidum*, (**B**) *Cryptococcus neoformans*, (**C**) *Rhodotorula mucilaginosa*, (**D**) *Prototheca zopfii*, (**E**) *Trichosporon asahii*, (**F**) *Candida parapsilosis*, (**G**) *Nakaseomyces glabrata*, (**H**) *Candida tropicalis*, (**I**) *Candida albicans*, (**J**) *Meyerozyma guilliermondii*, (**K**) *Malassezia pachydermatis*, (**L**) *Papiliotrema laurentii*, (**M**) *Cutaneotrichosporon jirovecii*, (**N**) *Trichosporon coremiiforme* and (**O**) *Debaryomyces nepalensis*.

**Table 1 jof-10-00218-t001:** Animal source and type of samples and classification of the cultures included in the study.

	Laboratory ID	Animal Source	Type of Sample	Microscopic Direct Examination Result	Culture Results
Domestic animals					
1	VS-397-22	Dog	Colon biopsy	Phagocytosed and free yeasts	Pure, significant
2	VS-207-23		Colon biopsy	Phagocytosed and free yeasts	Pure, significant
3	VS-057-23		Ear canal swab	Single-budding yeasts	Pure, significant
4	VS-063-23		Ear canal swab	Single-budding yeasts	Pure, significant
5	VS-520-22		Gastric lavage	Esporangia	Pure, significant
6	VS-428-22 (referred)		Prostate aspirate	No data	Pure, significant
7	VS-418-22		Skin scraping	No yeasts were observed	Pure, transitory
8	VS-056-23		Skin scraping	No yeasts were observed	Pure, transitory
9	VS-049-23		Skin scraping	No yeasts were observed	Pure, transitory
10	VS-475-21		Skin scraping	No yeasts were observed	Pure, transitory
11	VS-382-22		Skin scraping	No yeasts were observed	Pure, transitory
12	VS-347-22 (referred)		Skin of nasal plane biopsy	No data	Pure, significant
13	VS-076-21	Cat	Lymph node biopsy	Encapsulated budding yeasts	Pure, significant
14	VS-451-20 (referred)		Nasal biopsy	No data	Pure, significant
15	VS-002-23 (referred)		Nasal biopsy	No data	Pure, significant
16	3174-21E (referred)		Nasal lavage	No data	Pure, significant
17	4645-20E (referred)		Nasal lavage	No data	Pure, significant
18	VS-101-22		Skin nodule aspirate	Encapsulated budding yeasts	Pure, significant
19	4302-21S (referred)		Skin swab of an ulcerative lesion	No data	Mixed, two types of yeasts, significant
20	VS-515-22		Skin scraping	Budding yeasts	Pure, significant
21	VS-292-22 (referred)		Skin biopsy of a granular lesion	No data	Pure, significant
22	VS-013-23 (referred)		Submandibular node biopsy	No data	Pure, significant
23	INV-001-22		Urine for cryptococcosis monitoring	No yeast were obseved	Pure, significant
24	DOC-004-23	Domestic rabbit	Skin scraping	No yeast were obseved	Pure, transitory
25	VS-516-22 (referred)	Mare	Skin scraping	No data	Pure, significant
26	VS-257-22 (referred)		Uterine lavage	No data	Pure, significant
27	VS-211-23 (referred)		Uterine lavage	No data	Pure, significant
28	VS-234-23		Uterine lavage	Pseudomycelia and budding yeasts	Pure, significant
Exotic and Wild Animals					
29	BH64	Bat (*Pteronotus parnelli*)	Ear canal swab	No yeasts were observed	Pure, transitory
30	INV-006-22	Finsch’s parakeet (*Psittacara finschi*)	Lung biopsy	Not performed	Mixed, two types of yeasts, significance not determinated
31	INV-004-22	Mexican porcupine (*Sphiggurus* *mexicanus*)	Skin scraping	No yeasts were observed	Pure, transitory
32	INV-007-22	Northern tamandua (*Tamadua mexicana*)	Lung biopsy	Granulomatous inflammation with budding yeasts in the histopathology	Pure, significant
33	INV-009-22	Southern mealy parrot (*Amazona farinosa*)	Crop lavage	Budding yeasts	Mixed, three types of yeasts, significant
34	INV-001-20	Squirrel monkey (*Saimiri oerstedii*)	Kidney biopsy	No yeasts were observed	Pure, transitory
35	INV-005-22	Two-toed sloth (*Choloepus hoffmanii*)	Lung biopsy	Not performed	Pure, significance not determined
36	INV-011-22		Swabs of oral cavity lessions	Septate hyphae, pseudomycelia and budding yeasts	Pure, significant
37	INV-010-22		Swabs of oral cavity lesions	Budding yeasts	Mixed, three types of yeasts, significant
38	INV-002-23		Oral cavity and stomach biopsies	Non-budding yeasts	Pure, significant
39	INV-001-23		Skin scraping	No yeasts were observed	Pure, transitory

**Table 2 jof-10-00218-t002:** Results of the identification of the isolates by Brilliance^®^ *Candida*, Vitek^®^ 2 Compact and MALDI-TOF Biotyper^®^ MSP.

	Laboratory ID	Color in Brilliance^®^ *Candida* *	Vitek 2 Compact Identification	Probability (%)	MALDI-TOF Identification	Score	Observations
**1**	VS-418-22	Green	*Candida albicans*	96	*Candida albicans*	2.03	Full Agreement
**2**	INV-006-22	Green	*Candida albicans*	96	*Candida albicans*	1.95	Full Agreement
**3**	INV-011-22	Green	*Candida albicans*	99	*Candida albicans*	2.13	Full Agreement
**4**	INV-010-22	Green	*Candida albicans*	99	*Candida albicans*	2.15	Full Agreement
**5**	VS-382-22	Purple	*Candida ciferri*	98	*Meyerozyma guilliermondii*	1.88	Disagreement on genus and species
**6**	INV-002-23	Purple	*Candida guilliermondii*	50	*Meyerozyma guilliermondii*	2.31	Full agreement with non-acceptable probability
**7**	VS-056-23	Purple	*Candida guilliermondii*	98	*Meyerozyma guilliermondii*	2.41	Full Agreement with changes on nomenclature
**8**	DOC-004-23	Purple	*Candida guilliermondii*	98	*Meyerozyma guilliermondii*	2.09	Full Agreement with changes on nomenclature
**9**	INV-001-23	Purple	*Candida guilliermondii*	98	*Meyerozyma guilliermondii*	2.32	Full Agreement with changes on nomenclature
**10**	VS-049-23	Purple	*Candida guilliermondii*	98	*Meyerozyma guilliermondii*	2.32	Full Agreement with changes on nomenclature
**11**	BH64	Beige	*Candida parapsilosis*	97	*Candida parapsilosis*	2.08	Full Agreement
**12**	VS-428-22	Beige	*Candida parapsilosis*	97	*Candida parapsilosis*	2.13	Full Agreement
**13**	4302-21S	Beige	*Candida parapsilosis*	95	*Candida parapsilosis*	2.10	Full Agreement
**14**	VS-515-22	Beige	*Candida parapsilosis*	98	*Candida parapsilosis*	2.10	Full Agreement
**15**	VS-475-21	Beige	*Candida parapsilosis*	94	*Candida parapsilosis*	1.84	Full Agreement
**16**	VS-257-22	Beige	*Candida parapsilopsis*	98	*Candida orthopsilosis*	1.75	Partial Agreement, species complex
**17**	INV-007-22	Cream	*Candida glabrata*	99	*Nakaseomyces glabrata*	2,12	Full Agreement with changes on nomenclature
**18**	INV-009-22	Cream	*Candida glabrata*	99	*Nakaseomyces glabrata*	2.16	Full Agreement with changes on nomenclature
**19**	INV-010-22	Beige ****	*Candida famata*	86	*Debaryomyces nepalensis*	2.08	Disagreement on genus and species
**20**	INV-009-22	Dark blue	*Candida tropicalis*	98	*Candida tropicalis*	2.09	Full Agreement
**21**	VS-234-23	Dark blue	*Candida tropicalis*	99	*Candida tropicalis*	2.26	Full Agreement
**22**	INV-005-22	Dark blue	*Candida tropicalis*	99	*Candida tropicalis*	1.85	Full Agreement
**23**	VS-211-23	Dark blue	*Candida tropicalis*	96	*Candida tropicalis*	1.78	Full Agreement
**24**	INV-010-22	Greenish blue **	*Candida tropicalis*	91	*Cutaneotrichosporon* *jirovecii*	1.92	Disagreement on genus and species
**25**	INV-001-22	Beige	*Cryptococcus neoformans*	94	*Cryptococcus neoformans*	2.00	Full Agreement
**26**	VS-101-22	Beige	*Cryptococcus neoformans*	94	*Cryptococcus neoformans*	1.97	Full Agreement
**27**	VS-002-23	Beige	*Cryptococcus neoformans*	89	*Cryptococcus neoformans*	2.20	Full Agreement
**28**	4645-20E	Beige	*Cryptococcus neoformans*	92	*Cryptococcus neoformans*	2.00	Full Agreement
**29**	VS-292-22	Beige	*Cryptococcus neoformans*	93	*Cryptococcus neoformans*	2.26	Full Agreement
**30**	VS-076-21	Beige	*Cryptococcus neoformans*	95	*Cryptococcus neoformans*	2.03	Full Agreement
**31**	VS-013-23	Beige	*Cryptococcus neoformans*	96	*Cryptococcus neoformans*	2.07	Full Agreement
**32**	VS-451-20	Purple ****	*Cryptococcus laurentii*	90	*Papiliotrema laurentii*	2.30	Full Agreement with changes on nomenclature
**33**	VS-516-22	Purple **	*Geotrichum klebahnii*	93	*Geotrichum candidum*	2.10	Partial agreement on the genus
**34**	VS-057-23	Purple ***	*Malassezia pachydermatis*	97	*Malassezia pachydermatis*	2.20	Full Agreement
**35**	VS-063-23	Purple ***	*Malassezia pachydermatis*	99	*Malassezia pachydermatis*	1.26	Full Agreement but with non-acceptable score
**36**	VS-520-22	Light purple ****	*Prototheca zopfii*	96	No matched pattern	-	No Agreement could be done
**37**	VS-347-22	Light purple ****	*Prototheca wickerhamii*	93	No matched pattern	-	No Agreement could be done
**38**	INV-001-20	Pink	*Rhodotorula mucilaginosa/glutinis*	95	*Rhodotorula mucilaginosa*	2.28	Partial agreement on the genus
**39**	3174-21E	Pink	*Rhodotorula mucilaginosa/glutinis*	93	*Rhodotorula mucilaginosa*	2.08	Partial agreement on the genus
**40**	4302-21S	Pink	*Rhodotorula mucilaginosa/glutinis*	89	*Rhodotorula mucilaginosa*	2.10	Partial agreement on the genus
**41**	VS-207-23	Greenish blue	*Trichosporon* sp.	NA	*Trichosporon asahii*	2.19	Partial agreement on the genus
**42**	VS-397-22	Greenish blue	*Trichosporon asahii*	90	*Trichosporon asahii*	2.09	Full Agreement
**43**	INV-006-22	Greenish blue	*Trichosporon asahii*	99	*Trichosporon asahii*	2.18	Full Agreement
**44**	INV-009-22	Greenish blue	*Trichosporon asahii*	94	*Trichosporon coremiiforme*	2.26	Partial agreement on the genus
**45**	INV-004-22	Greenish blue	*Trichosporon* sp.	NA	*Trichosporon coremiiforme*	2.14	Partial agreement on the genus

* The colors of the isolates obtained coincided with what was described [12,13,14]. ** For those species, no data were found in the consulted references [12,13,14], but the colors coincided with the ones presented by other members of the same or related genera. *** *Malassezia* spp. have their one chromogenic media from the same brand that we used, and in that media, the developed color was the same as the one obtained in this research. **** No data were found in the consulted references [12,13,14] to compare to the obtained colors.

**Table 3 jof-10-00218-t003:** Distribution of the isolates according to animal source, sampled organ and significance.

Isolate (*n* > 1)	Animal Source (*n* > 1)	Sampled Organ	Classification of the Isolate (*n* > 1)
*Candida albicans* (4)	Dog	Skin	Transitory
	Finsch’s parakeet	Lung	Non-determined
	Two-toed sloth (2)	Mouth	Significative (2)
*Meyerozyma guilliermondii* (6)	Dog (3)	Skin	Transitory (3)
	Rabbit	Skin	Transitory
	Two-toed sloth (2)	Skin	Transitory
		Mouth	Significative
*Candida parapsilosis* (5)	Dog (2)	Prostate	Significative
		Skin	Transitory
	Cat (2)	Skin	Significative (2)
	Bat	Ear canal	Transitory
*Candida orthopsilosis*	Mare	Uterus	Significative
*Nakaseomyces glabrata* (2)	Northern Tamandua	Lung	Significative
	Southern mealy parrot	Crop	Significative
*Debaryomyces nepalensis*	Two-toed sloth	Mouth	Significative
*Candida tropicalis* (4)	Mare (2)	Uterus	Significative (2)
	Southern mealy parrot	Crop	Significative
	Two-toed sloth	Lung	Non-determined
*Cutaneotrichosporon jirovecii*	Two-toed sloth	Mouth	Significative
*Cryptococcus neoformans* (7)	Cat	Nose (2)	Significative
		Skin (2)	Significative
		Lymph node (2)	Significative
		Bladder	Significative
*Papiliotrema laurentii*	Cat	Nose	Significative
*Geotrichum candidum*	Mare	Skin	Significative
*Malassezia pachydermatis* (2)	Dog	Ear canal	Significative
*Prototheca zopfii*	Dog	Stomach	Significative
*Prototheca wickerhamii*	Dog	Nose	Significative
*Rhodotorula mucilaginosa* (3)	Cat (2)	Nose	Significative
		Skin	Significative
	Squirrel monkey	Kidney	Transitory
*Trichosporon asahii* (3)	Dog (2)	Colon	Significative
	Finsch’s parakeet	Lung	Non-determined
*Trichosporon coremiiforme* (2)	Mexican porcupine	Skin	Transitory
	Southern mealy parrot	Crop	Significative

## Data Availability

All the data were included in this manuscript.

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
