# Peer review of "Chromogenic, Biochemical and Proteomic Identification of Yeast and Yeast-like Microorganisms Isolated from Clinical Samples from Animals of Costa Rica"

_jof, 2024, doi:10.3390/jof10030218_

Round 1

Reviewer 1 Report

I consider this an important study, since there is not much information available about the occurrence of fungi of medical importance in animals, especially in wild animals. This may be the beginning for health and veterinary entities to take the necessary measures to deal with this problem.

I consider that the article is well written, the introduction mentions the most relevant information to address everything that was done. The methodology was the right one to fulfill the main objective of the work, however, I still do not understand the purpose of using gram staining, since there are other stains that are more specific for fungi. This part should be explained in detail. 

As for the results, the tables and figures seem good to me, they contain the necessary information for the reader to understand which species of fungus was isolated from each animal, both domestic and wild. 

In figure 3, it seems to me that the photos are not the best, if possible improve them, and put size scales.

I would like the discussion to focus a little more on the importance of these studies, bearing in mind that most of the fungal species reported here are pathogenic to humans. How did these pathogens get into wild animals? To pose a hypothesis of how they appeared.

in the attached file I mention the most relevant comments that should be addressed. 

Author Response

Dear Reviewer: 

Thank you for taking the time to read and make valuable comments on our work, these were the suggested changes and our responses:

  1. Explain the choosing of Gram Stain in this study instead of using other more specific for mycology. Response: we use Gram Stain for two reasons; to detect bacterial contamination of the isolates and because is the most used in clinical laboratories where mycology is done in the same facilities as bacteriology, therefore we thought it would be useful for readers to see the images with that staining. However, as the reviewer points out, there is some other stains more specific, like Methylene Blue, we tried to take new pictures with that stain for this new version of the paper, however some strains needs to be subculture at least three times to recover its vitality, and there is not enough time to this because the journal give us one week to resend the article. For that reason, we explain the choosing of Gram Stain as follows in lines 103-104: “Gram stain was chosen to detect and classified bacterial contamination of the isolates.”
  2. Improve the quality of figure 3 and put size bars. Response: since we have not enough time to subculture the strains for this new version and to take photography with a new stain, we improve the color of the images of figure 3 and put a scale bar more evident, since the red one that we used in the former version was very thin, also, we indicate that the scale of the bar is 10.33 µm in the legend of that figure (line 217).
  3. I would like the discussion to focus a little more on the importance of these studies, bearing in mind that most of the fungal species reported here are pathogenic to humans. How did these pathogens get into wild animals? To pose a hypothesis of how they appeared”. Response: to accomplish this comment we include two paragraphs, one just after the sentence about the Fungal Pathogens List about human cases of candidemia and cryptococcosis in Costa Rica, as follows in lines 237-248: “Considering that, in Costa Rica, there is few reports on yeasts infections also in humans, being the majority retrospective studies of candidemia and cryptococcosis. Regarding candidemia, INCIENSA has the records from the period 2018-2021 of the blood cultures from the main hospitals, being albicans, C. parapsilosis, C. tropicalis and N. glabrata (in that order) the most frequently identified [20]. However, in the two largest hospitals of this country (Hospital San Juan de Dios and Hospital México), the order of the fungal pathogens varies between places and periods [21, 22, 23]. In reference to cryptococcosis, this disease is prevalent in the country, mainly in patients with Human Immunodeficiency Virus (HIV) (72.7%), but also in immunocompetent patients (20%) and the rest by drug-mediated immunodeficiencies [24].  Being all these fungal pathogens transmitted by exposure to environment or by nosocomial and medical devices contamination, that reinforces the necessity of collaboration between human and animal diagnoses to detect mutual sanitary threats.” The other paragraph was included between lines 324-338 and it said: “Although reports of yeasts in wild animals are very scarce, their transient isolation in those kind of animals in the present study is not surprising, since, according to a review on evolution and genetics of yeasts, of the 76 new species described recently, only seven were related with human environments, been the other sources soil, plants, fruits, water, and insects, among others [42]; and those other sources are more likely to get in touch (by contact or by ingestion) with wild animals than humans. Also, yeast pass the gastrointestinal tract and could be isolate from feces, like in the survey of 325 droppings of parrots in Italy, where 49.2% from those were positives to 27 different species of yeasts, been the majority from the genus Candida, but also from genus Debaryomyces, Geotrichum, Pichia and Rhodotorula [43]. Other example is the research on the mycobiome of wild herbivorous mammals (genus Neotoma) in 25 populations in southwestern United States, in which fungal assemblages were dominated by plant pathogens and molds, but its composition varied by animal’s habitat and significantly decrease in captivity [44]. Therefore, in the cases where significant isolation of yeasts was detected in this study, it could be attributed to the stress of captivity and antibiotic therapy to which animals are exposed.”  
  4. Improve keywords. Response: the keywords of the new manuscript are the following (lines 32-33): proteomic identification; pathogenic fungi; biochemical profile; chromogenic media; Prototheca; wild animals; domestic animals; veterinary mycoses; One Health.
  5. Change the word onerous. Response: the word was changed to expensive (line 56).
  6. In table 1, when more than one case of the same animal species where indicated, write only the first one. Response: when several cases of the same animal species where indicated, only in the first line the species where indicated. For example: dog, cat, mare, among others.
  7. In table 2, write Candida in italics after the brand Brilliance. Response: italics were added to the word Candida before the brand Brilliance (line 182 and label of Table 2).
  8. In the legend of table 3, add italics to the scientific names. Response: italics were added to the scientific names.
  9. In the discussion section, to what can the presence of Cryptococcus neoformans in cats be attributed? Response: this paragraph were added in that part of the discussion in lines 255-267: “Due to the behavior of burying feces and hunting, cats are exposed to acquiring fungal agents from the environment, which can cause disease or remaining on their surfaces as asymptomatic carriers, which is why in the case of sapronosis (fungal infections acquired from the environment) they are good bioindicators of contaminated environments [27] and, could indicate a common source of exposure of the people they live with. Cryptococcosis is the most common systemic fungal disease in domestic felines, and manifests mainly as an inflammation of the nasal mucosa that may or may not spread to the lower respiratory tract, and causing lymphadenopathy, or infection of the central nervous system [27]. Contrary to what happens in humans, in which the disease caused by C. neoformans manifests mainly in immunosuppressed patients [18], in cats it has been seen that this disease is also contracted by animals in which immunosuppression was not detected [27], opening the possibility that a large inoculum from the environment can promote the development of signs.”

Reviewer 2 Report

Is a relevant work because it shows the identification of yeasts and yeast-like microorganisms from domestic and wild animals. Currently, there are few references on fungal microorganisms in animals.

No comments

Author Response

Thank you for the review on our work.

Reviewer 3 Report

Dear Authors

I consider the article to be very interesting and complete.

I would just like to make a few comments. See in detail comments

Best regards

1. What do you mean by "nervous commitment"? Does it mean that it affects the central nervous system?

2. In materials and methods, identification by MALDI-TOF, it says in lines 128 and 129: 1 L and it should say 1 microliter (1 µL).

3. Please clarify what do you mean with “significant, transitory or undetermined” (line 145).

4. Tables 2 and 3 correct jirovecii (it says jirovecci). Also, on line 306

5. Legends of figure 3. (lines 211 to 216) Place the names of the microorganisms in italics.

Author Response

Dear Reviewer: 

Thank you for taking the time to read and make comments and make valuable comments on our work, these were the suggested changes and our responses:

  1. What do you mean by "nervous commitment"? Does it mean that it affects the central nervous system? Response: in line 49, the sentence was corrected as follows: “…and central nervous system compromise”
  2. In materials and methods, identification by MALDI-TOF, it says in lines 128 and 129: 1 L and it should say 1 microliter (1 µL). Response: in line 133 and 134, the recommended correction was done.
  3. Please clarify what do you mean with “significant, transitory or undetermined” (line 145). Response: lines 92 to 101 specifies the meaning of those terms as follows: “Significant growth was considered when: 1) colonies appeared over the inoculated sample in case of biopsies and scrapings, in at least one zone of growth using the streak method and when it was presence of encapsulated, budding or phagocytized yeasts or hyphae and pseudomycelia in the direct examination (Gram and/or Giemsa stains); 2) when the isolation came from a sterile organ; 3) when was a culture of treatment monitoring of a previous diagnosed disease; and, 4) when it was a isolate submitted for identification from private veterinary diagnostic laboratory. Transitory growth was annotated when no fungal elements were seen in the direct examination. The significance of growth where not determined in the cases where no direct examination was made to classify it as significant or transitory.” Then in lines 149-150, the phrase now says: “(significant, transitory or significance not determined, as explained above).
  1. Tables 2 and 3 correct jirovecii (it says jirovecci). Also, on line 306. Response: the correction was made in both tables and in line 349.
  2. Legends of figure 3. (lines 211 to 216) Place the names of the microorganisms in italics. Response: the names of the microorganisms in figure 3 were placed in italics (lines 217-221).